# Multidimensional Impact of Urbanization Process on Regional Net CO₂ Emissions: Taking the Yangtze River Economic Belt as an Example

**Xiaomei Shen [1,†], Hong Zheng [2,†], Mingdong Jiang [3,*,†], Xinxin Yu [1,4], Heyichen Xu [5] and Guanyu Zhong [6]**

1. Green, Low-Carbon and Circular Economy Institute of Yancheng, Yancheng Institute of Technology, Yancheng 224051, China; sxm@ycit.cn (X.S.); 21s016011@stu.edu.cn (X.Y.)
2. School of Business, Hohai University, Nanjing 211100, China; hhuzh2022@163.com
3. College of Environmental Sciences and Engineering, Peking University, Beijing 100871, China
4. School of Humanities, Social Sciences and Law, Harbin Institute of Technology, Harbin 150001, China
5. School of Environment, Hohai University, Nanjing 210098, China; xhyc13656208555@163.com
6. UNSW Business School, University of New South Wales, Sydney, NSW 2052, Australia; zgy9734@outlook.com
* Correspondence: pkujmd@126.com; Tel.: +86-13-436-388-706
† These authors contributed equally to this work.

**Abstract:** Urbanization is a powerful symbol and an inevitable human economic and social development trend. This process affects carbon dioxide emissions by changing the human output and lifestyle and encroaches over the carbon sink areas by adjusting the land use types impacting the regional carbon balance. We systematically analyzed the influence of urbanization on regional net CO₂ emissions (NCE) and built a quantitative model for the impact of urbanization on NCE based on population, economy, and land use. Based on this, the Yangtze River Economic Belt (YREB) in China has been selected as an example to measure the characteristics of the spatial and temporal evolution of NCE from 2005 to 2018 by empirically testing the contributions of population urbanization, economic urbanization, and land urbanization to the NCE changes in YREB. According to the study's findings, the carbon-neutral pressure index of the YREB increased over the study period, with an increase in NCE from 1706.50 Mt to 3106.05 Mt. The contribution of urbanization in this process increased and subsequently decreased in an inverted U pattern with a drop in the cumulative net emission of 260.32 Mt. The inflection points of the cumulative impact of urbanization on NCE in the midstream and upstream regions occurred in 2011 and 2010, respectively. Due to the high degree of urbanization and economic growth in the downstream area, the urbanization impact demonstrated a constant reduction of NCE over the research period. In terms of sub-dimensions, the population and land urbanization effects were consistently positive, while the economic urbanization affected the NCE and displayed an inverted U pattern during the study period. If the variation in regional carbon sink space is ignored, the impact of urbanization on CO₂ emission reduction will be overestimated. We investigated the realization path of differentiated synergistic emission reduction strategies in the great river economic belts based on the empirical study on YREB.

**Keywords:** the net CO₂ emissions; urbanization; multidimensional impacts; the Yangtze River Economic Belt; carbon neutralization

## 1. Introduction

Global warming induced by greenhouse gas emissions has received global attention in recent years. Carbon dioxide is the most significant contributor to climate change among greenhouse gases emitted by human activity, accounting for 63% of the total warming effect of all greenhouse gases. Being the second-largest economy globally and the nation with the highest CO₂ emissions, China is under enormous pressure to reduce emissions [1,2]. According to a crude development pattern, China achieves economic growth by consuming

large amounts of energy, resulting in a rapid increase in carbon emissions [3], and the most important driving factor exacerbating this situation is rapid urbanization [4]. The urbanization rate of China has increased dramatically over the last four decades, from 17.9% in 1978 to 60.6% in 2019 [5], and the urban population of the country is expected to surpass 75% of the overall population by 2050 [6]. Urbanization is accompanied by industrial economic growth, increasing energy consumption, lifestyle changes, and changes in land use types due to economic and social progress, all of which have a significant influence on carbon emissions [7]. The Chinese government urged China at the 75th United Nations General Assembly in September 2020 to increase its efforts to reduce emissions on its own and strive to achieve peak carbon dioxide emissions by 2030. In this context, exploring the low-carbon development path under the rising urbanization has become a critical concern for the Chinese government.

Some previous research on the impact of urbanization on carbon emission effects has reported increased carbon emissions in response to progressing urbanization. The STIRPAT model using data from 17 developed economies between 1960 and 2005 by Liddle and Lung [8] suggested an increase in residential energy consumption due to urbanization, resulting in increments in carbon emissions. The positive influence of urbanization on carbon emissions in BRICS countries conducted by Wang et al. [9] using the data from 1985 to 2014 reported significant exacerbations in carbon emissions. Adusah-Poku [10] argued that the urbanization caused an increase in $CO_2$ emissions both in the long and short run using panel data from 1990–2010 and a newly established pooled mean group (PMG) estimator. Similarly, the autoregressive distributed lag model and the vector error correction model built by Ali et al. [11] to investigate the relationship between urbanization and carbon emissions in Pakistan from 1972 to 2014 showed the significant contribution of impending urbanization on $CO_2$ emissions. The direct and indirect $CO_2$ emissions from Chinese households from 1996 to 2012 estimated by Li et al. [12] using the input-output method showed an increase by 2.9% and 1.1%, respectively, for every 1% increase in urbanization level. However, the relationship between urbanization and carbon emissions was reported to be complicated by many scholars. For example, Kocoglu et al. [13] studied the relationship between urbanization and carbon emissions in 15 emerging economies from 1995 to 2015 using threshold analysis and the quantile regression method and found an inverted U-shaped effect of urbanization on carbon emissions. Also, Khan and Su [14] reported that when the level of urbanization was below the threshold, the carbon emissions in new industrial countries were urbanization-induced, and when the urbanization exceeds the threshold, carbon emissions are moderated. Chen et al. [15] also observed an inverted U-shaped relationship between urbanization and $CO_2$ emissions in western China based on data from 188 prefecture-level cities in China. The investigation of the relationship between urbanization and carbon emissions by Fan and Zhou [16] using the M-LMDI model and spatial panel data model in 30 Chinese provinces from 1997 to 2015 proved the negative impact of urbanization on carbon emissions and the positive spatial spillover effect. Shi and Li [17] analyzed the dynamic relationship between urbanization rate and carbon emissions in different urban agglomerations using a spatial autoregressive model and established a three-stage dynamic relationship between urbanization rate and carbon emissions that differed among urban agglomerations. Furthermore, the nonlinear relationship between urbanization and carbon emissions was also documented by Guo et al. [18], Hashmi et al. [19], and Muhammad et al. [20].

Several studies on the mechanism of the impact of urbanization on $CO_2$ emissions have attempted to explain the spatial and temporal heterogeneity. As a whole, urbanization affects carbon emissions through population and economy. As for population urbanization, the major manifestation is the population migration from rural to urban regions, which is also the main focus of prior research on the mechanism of the carbon emission effect of urbanization [21–23]. The urbanization process drives the change in the lifestyle of the inhabitants. The demand of residents for housing and public facilities has been consistently increasing with urbanization, driving up the consumption of high-emission products such

as steel and construction materials [12,24]. At the same time, the increased usage of motor vehicles and electrical and electronic equipment further raise the energy demand [25]. In contrast, the supplement of electricity, heat, and energy in the city tends to achieve intensification, high efficiency, and low carbonation. With the advancement of urbanization, electricity, natural gas, and liquefied petroleum gas have replaced traditional coal cakes on the consumer side. Furthermore, on the energy supply side, the clean-up of a consistent urban power supply system is also more cost-effective than upgrading the rural distribution facilities. Furthermore, the urban residents are better educated and more inclined to understand and embrace low-carbon living. Low-carbon transport and consumption are becoming a new urban fashion [26]. Except for changes in emissions from domestic sources, the migration of the population from rural and urban areas results in increased productivity and subsequent changes in regional industrial structures. The economic urbanization has greater effects on regional carbon emissions than demographic changes [27,28]. During the early stage of urbanization, the secondary industry expands quickly, resulting in a significant consumption of fossil energy and mineral resources as the primary source of $CO_2$ by industries and constructions that is accompanied by a substantial increase in $CO_2$ emissions [29]. With the continuous progress of urbanization, the secondary industry is gradually evolving into the tertiary industry. Therefore, the banking, commerce, and other service functions continue to improve. The value realization process of the tertiary industry is less dependent on resource consumption and discharges less pollutant emissions than the material-production-based primary and secondary industries, so the impact of economic urbanization on carbon emissions also has positive effect [28].

Besides, it is worth noting that the urbanization process has a complex impact on carbon emissions and changing regional $CO_2$ absorption levels due to structural changes in land use types, which has an impact on the regional carbon balance. Carbon dioxide absorption by surface vegetation is an essential service function of ecosystems that plays a critical role in the global carbon cycle and climate control. However, in the process of urbanization, a large number of woodlands, grasslands, and wetlands are occupied and transferred for construction uses [30,31]. China's urban construction land increased from 168,015 $km^2$ to 215,841 $km^2$ between 2000 and 2015, with a rise rate of 28.47% [32]. The continuous conversion of large-scale natural ecosystems such as forests, grasslands, and wetlands to arable land has resulted in a significant reduction in the level of regional carbon sinks to compensate for the massive expansion of urban areas and the subsequent loss of farmlands [33]. Wang et al. [34] observed a decline in carbon stocks due to rapid urbanization in the Beijing-Tianjin-Hebei region of China from 4.055 $\times$ $10^8$ t in 1990 to 4.004 $\times$ $10^8$ t in 2015. Between 2009 and 2019, urbanization resulted in a 1.75% reduction in vegetation cover in Cumilla, Bangladesh, and the loss of carbon sink function resulted in an eight degrees Celsius rise in surface temperature in the study area [35]. We can conclude that the impact of land urbanization on the condition of regional carbon balance should not be overlooked.

In conclusion, despite many research attempts to investigate the influence of urbanization on carbon emissions, there is no consensus on whether this impact is beneficial or detrimental. Scholars have attempted to analyze the causes of these disparities from several perspectives, including the urban-rural structure of the population and the economic structure. However, only a few studies have included both in the same analytical framework [36,37]. It is noteworthy that the existing literature only focuses on the impact of urbanization on regional carbon balance from the perspective of $CO_2$ emission sources. Carbon emissions are not only the sole factor influencing the global greenhouse gas concentrations, but the carbon absorption capacity of regional ecosystems is equally essential. Urbanization, as one of the key driving forces behind global economic and social development [38], affects the regional carbon emissions by changing the economic production structures and residents' lifestyles, as well as the regional ecological carbon sink capacity by changing the regional ecological carbon sink capacity of the land use types. The impact

of urbanization on regional net carbon emissions is more important to research than the carbon emissions that have been the focus of the previous studies.

As the most influential inland river economic belt in the world, the Yangtze River Economic Belt (YREB), with over 40% of the national population and economy, has an extremely important strategic and ecological position in China [39]. The 11 provinces and cities that make up the YREB can appropriately reflect the urbanization and net $CO_2$ emission characteristics of areas with varying degrees of economic growth, since it encompasses three regions in China's east, centre, and west. As a result, research into the influence of urbanization on net $CO_2$ emissions in the YREB serves as a model for other Chinese areas and even underdeveloped nations. In this context, this study monitors and analyzes the $CO_2$ emissions from human activities and $CO_2$ absorption by the ecosystem in each YREB province from 2005 to 2018. Furthermore, the spatial and temporal characteristics of the net carbon emissions values (*NCE*), per capita net emissions (PNE), and carbon-neutral stress index (CSI) in the YREB are also investigated. In addition, the spatial and temporal heterogeneity and patterns of urbanization on net carbon emissions in different regions are assessed by tracing the contributions of population urbanization, economic urbanization, and land urbanization to changes in *NCE* in the YREB, respectively, using the LMDI model. The marginal contributions of this paper are as follows. Firstly, we place carbon sources and sinks in the same research framework to explore the way urbanization affects net regional $CO_2$ emissions. Secondly, we build a framework to analyze the impact of urbanization on net $CO_2$ emissions in three dimensions: population, economy, and land, and further introduce the LMDI model to quantify the effects. Thirdly, due to the gap in urbanization levels in different regions of the YREB, we further analyze the spatial and temporal heterogeneity and patterns of the impact of urbanization on net $CO_2$ emissions.

## 2. Methods and Data

### 2.1. Methodology

#### 2.1.1. Calculation of the *NCE*

The *NCE* refers to the carbon emission minus carbon sequestration in a region, which can directly reflect the low-carbon level of the region and the difficulty of carbon neutralization. The higher the *NCE* of a region, the lower its low-carbon level is. In addition, it would be more difficult to achieve carbon neutralization. The calculation method of *NCE* is as follows:

$$NCE = CE - CA \tag{1}$$

where $CE$ denotes the value of carbon emissions, and $CA$ is the value of carbon sequestration. Furthermore, we decompose the total carbon emissions in the YREB into carbon emissions from the production sector and the living sector [40,41]. Among them, the living sector includes urban and rural areas, and the production sector includes primary, secondary and tertiary industries. For energy consumption, we involve 17 energy species, while for the calculation of carbon sequestration, we divide the carbon sink into 23 land types. The decomposition models of carbon emission and carbon sequestration are shown in Equations (2) and (3):

$$CE = C_P + C_L = \sum_{i=1}^{5} C_i \tag{2}$$

$$CA = \sum_{i=6}^{28} C_i \tag{3}$$

where subscript *i* represents different production and living sectors. Among them, *i* = 1, 2, 3 denotes the production sector, including primary industry, secondary industry and tertiary industry; *i* = 4, 5 denotes the living sector, including urban and rural areas; and *i* = 6, 7,..., 28 denotes different land use types. Additionally, $C_i$ represents the carbon emission

generated by the sector $i$, $C_P$ represents the carbon emissions of the production sector, and $C_L$ represents the carbon emissions from the living sector.

### 2.1.2. The LMDI Method

To explore the effect of urbanization from the many factors affecting net emissions, we decompose the net emission values of the YREB using the LMDI model and extract the urbanization effect from them. The LMDI decomposition method was first proposed by Ang [42]. Since this decomposition method can achieve the complete decomposition of factors, truly achieving full decomposition and with no residuals, it is often considered as the optimal decomposition model and is widely utilized in the fields of energy and carbon accounting. The decomposition formula of *NCE* constructed according to the LMDI model is shown in Equation (4):

$$
\begin{aligned}
NCE = C_P + C_L - CA &= \sum_{i=1}^{3} C_i + \sum_{i=4}^{5} C_i - \sum_{i=6}^{28} C_i \\
&= \sum_{i=1}^{3} IE_i \times IS_i \times IU_i \times IC_i + \sum_{i=4}^{5} LE_i \times LS_i \times LU_i \times LC_i + \sum_{i=6}^{28} DE_i \times DS_i \times DU_i \times DC_i \\
&= \sum_{i=1}^{3} \frac{C_i}{E_i} \times \frac{E_i}{GDP_i} \times \frac{GDP_i}{GDP} \times GDP + \sum_{i=4}^{5} \frac{C_i}{E_i} \times \frac{E_i}{P_i} \times \frac{P_i}{P} \times P + \sum_{i=6}^{28} \frac{C_i}{S_i} \times \frac{S_i}{S} \times \frac{S}{Size} \times Size
\end{aligned}
\tag{4}
$$

where $E_i$ represents the total energy consumption of the sector $i$, $GDP_i$ is the output of the production sector $i$, $GDP$ is the Gross Regional Product, $P_i$ denotes the total population in the living sector $i$, $P$ denotes the total population of a region, $S_i$ is the area of the land use type $i$, $S$ is the area of non-constructive land and $Size$ is the total area of a region. We use the LMDI model to decompose the *NCE* into values of 12 different effects. The designations of 12 effects and their meanings are shown in Table 1.

**Table 1.** The designations of 12 effects.

| | Designation | Abbreviation | Meaning |
|---|---|---|---|
| The economic effects | the economic emission effect | $IE_i$ | the carbon emission generated by per unit energy consumption of the production sector $i$ |
| | the economic strength effect | $IS_i$ | the energy consumption generated by per unit economic output of the production sector $i$ |
| | the economic urbanization effect | $IU_i$ | the proportion of output in the total production sector $i$ |
| | the economic scale effect | $IC_i$ | the total regional output value |
| The population effects | the population emission effect | $LE_i$ | the carbon emissions generated by per unit of energy consumption in the living sector $i$ |
| | the population strength effect | $LS_i$ | the energy consumption generated by per 10,000 people in the living sector $i$ |
| | the population urbanization effect | $LU_i$ | the proportion of the population in the living sector $i$ to the total population of the region |
| | the population scale effect | $LC_i$ | the total population of a region |
| The land effects | the land emission effect | $DE_i$ | the ratio of carbon sequestration of land type $i$ to its area |
| | the land strength effect | $DS_i$ | the ratio of land type $i$ to non-constructive land area |
| | the land urbanization effect | $DU_i$ | the ratio of non-constructive land to the area of the region |
| | the land scale effect | $DC_i$ | the total area of the region |

We assume that the time changes from year $t$-1 to year $t$. The year $t$-1 is the base year and year $t$ is the target year. $NCE_{t-1}$ and $NCE_t$ represent carbon emissions in year $t$-1 and year $t$. In the case of the additive decomposition framework, the *NCE* additives ($\Delta NCE$) from year $t$-1 to year $t$ is equal to the cumulative value of each effect. The decomposition of *NCE* additives into their component parts is shown in Equation (5):

$$\Delta NCE = NCE_t - NCE_{t-1} = \Delta IE + \Delta IS + \Delta IU + \Delta IC + \Delta LE + \Delta LS + \Delta LU + \Delta LC - \Delta DE - \Delta DS - \Delta DU - \Delta DC \quad (5)$$

where $\Delta IE, \Delta IS, \Delta IU, \Delta IC, \Delta LE, \Delta LS, \Delta LU, \Delta LC, \Delta DE, \Delta DS, \Delta DU$ and $\Delta DC$ represent each effect value respectively, and the decomposition value obtained represents the contribution value to the change of *NCE*.

We extract the economic urbanization effect value $\Delta IU$, the population urbanization effect value $\Delta LU$, and the land urbanization effect value $\Delta DU$ from the decomposition results based on three ways affecting *NCE* in the urbanization process, including industrial structure upgrading, urban-rural population share change, and land structure adjustment. Among them, economic urbanization is the proportion of output value of each industry to total output value, population urbanization is the proportion of each living sector to total population and land urbanization is the proportion of non-constructive land to total area of the region. The calculation formula for carbon emissions from urbanization effects in three dimensions are shown in Equations (6)–(8):

$$\Delta IU = \sum_{i=1}^{3} \frac{NCE_{i,t} - NCE_{i,t-1}}{\ln NCE_{i,t} - \ln NCE_{i,t-1}} \times \ln \frac{IU_{i,t}}{IU_{i,t-1}} \quad (6)$$

$$\Delta LU = \sum_{i=4}^{5} \frac{NCE_{i,t} - NCE_{i,t-1}}{\ln NCE_{i,t} - \ln NCE_{i,t-1}} \times \ln \frac{LU_{i,t}}{LU_{i,t-1}} \quad (7)$$

$$\Delta DU = \sum_{i=6}^{28} \frac{NCE_{i,t} - NCE_{i,t-1}}{\ln NCE_{i,t} - \ln NCE_{i,t-1}} \times \ln \frac{DU_{i,t}}{DU_{i,t-1}} \quad (8)$$

$NCE_{i,t-1}$ and $NCE_{i,t}$ represent the net carbon emission value of sector $i$ in year $t$-1 and year $t$ respectively. In addition, $\Delta TOT$ is the urbanization effect, which is the joint effect of economic urbanization effect, population urbanization effect and land urbanization effect. Its calculation method is as follows:

$$\Delta TOT = \Delta IU + \Delta LU - \Delta DU \quad (9)$$

*2.2. Data Sources*

This study uses the population, economic, energy, carbon emission, and carbon absorption data of 11 provinces and cities in the YREB from 2005–2018. Among them, the data related to population and economy are obtained from the China Statistical Yearbook [43], mainly including the year-end population, urban and rural share of the total regional population, regional GDP, and the share of output value of each industry in the annual total output value of each province. In particular, we have deflated the economic data based on 2005. The energy and carbon emission data are mainly from the Carbon Emission Accounts & Datasets (CEADs) [44], which contains the consumption information of 17 types of fossil fuels and carbon emissions from energy consumption of 47 economic sectors. Among them, the consumption of each type of energy is converted into standard coal according to the conversion coefficient of standard coal in the China Energy Statistics Yearbook [45]. For the carbon sequestration data, the area and NDVI coefficients of each land use type in the YREB were obtained from the database of the Chinese Academy of Sciences, and land use type and corresponding carbon sink coefficient are obtained referring to Zhang et al. [46]. Some data were calculated based on existing data. Limited by the availability of data, the research period involved in this study is from 2005 to 2018.

**3. Results**

*3.1. Analysis of Emission Reduction Pressure in Different Regions*

Figure 1 shows that the *NCE* of the YREB increased from 1706.50 Mt in 2005 to 3106.05 Mt in 2018, indicating an increasing trend from 2015 to 2018, reflecting the increasing

pressure to reduce emissions in the YREB. However, the government of China increased attention to emission issues and slowed down the growth in *NCE* significantly after 2011, since China's 12th Five-Year Plan, when the government established explicit emission reduction targets and introduced more emission reduction measures. From a regional perspective, the *NCE* was higher in the provinces of the downstream economic belts such as Jiangsu, Zhejiang, and Anhui, among which the *NCE* of Jiangsu province had been at the top during the study period, reaching 717.54 Mt in 2018, while the *NCE* of Chongqing, Yunnan and other regions in the upstream economic belt was relatively low, indicating substantial difficulty in achieving the carbon neutrality and severe emission reduction in regions with high development levels.

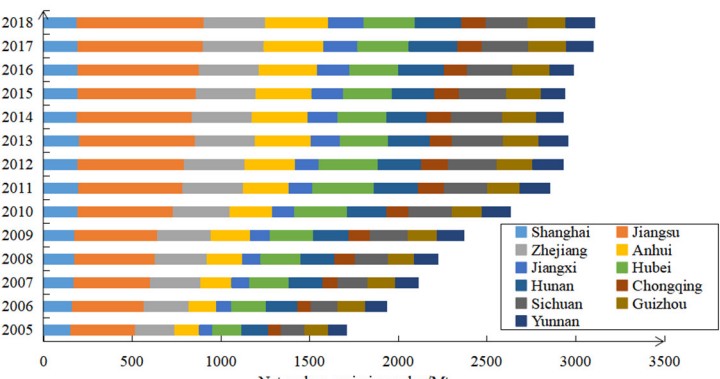

**Figure 1.** Net carbon emissions of 11 provinces and cities in the YREB from 2005 to 2018 [source: own illustration].

In addition to *NCE*, we chose another two indicators, the carbon neutral stress index (the ratio of carbon emissions to carbon sequestration by provinces and cities, CSI) and the net per capita emission value (PNE), to further measure the net emission status and the difficulty of achieving carbon neutrality in 11 provinces and cities in the YREB (Figure 2). The region with the largest CSI within 2005–2018 was Shanghai, whose values were above 600 during the study period, and even reached 955.66 in 2015, indicating that Shanghai's *NCE* were seriously overspent and it was hard to achieve carbon neutrality [47,48]. The CSI of Jiangsu Province was second only to that of Shanghai, and had a tendency to increase year after year. The provinces with lower CSI in the Yangtze River Delta urban agglomeration of the lower regions were Zhejiang Province and Anhui Province, and the rise of CSI in Anhui Province was even greater, which even surpassed Zhejiang Province in 2018, ranking third in the Yangtze River Economic Belt. This might be due to the strong reliance on traditional energy sources, such as coal, for the overall development of Anhui Province, which has led to an increase in carbon emissions [49]. Compared to the Yangtze River Delta urban agglomeration of the lower regions, the CSI of the rest of the provinces and cities of YREB were lower, basically below 30. The highest PNE was still found in Shanghai and Jiangsu Province. In 2005–2013 Shanghai was the city with the top PNE, and Jiangsu Province was the second, but Jiangsu surpassed Shanghai in 2014 and maintained the top position in the YREB during 2014–2018. Among the remaining provinces, the PNE values are relatively high in Zhejiang, Hubei, Anhui and Guizhou. From the change trend, the PNE of most provinces and cities showed a certain upward trend, especially in Jiangsu, Anhui and Jiangxi, and only Shanghai PNE values showed a slight downward trend, indicating the increasing pressure of emission reduction in the YREB as a whole. In a comprehensive analysis, the ranking of the 11 provinces and cities in the YREB was relatively close on the two indicators of CSI and PNE, which reflected that the greatest pressure on carbon emission reduction was in the economically developed downstream regions such as Shanghai and Jiangsu, while the least pressure on emission reduction was in the upstream regions such as Sichuan and Yunnan.

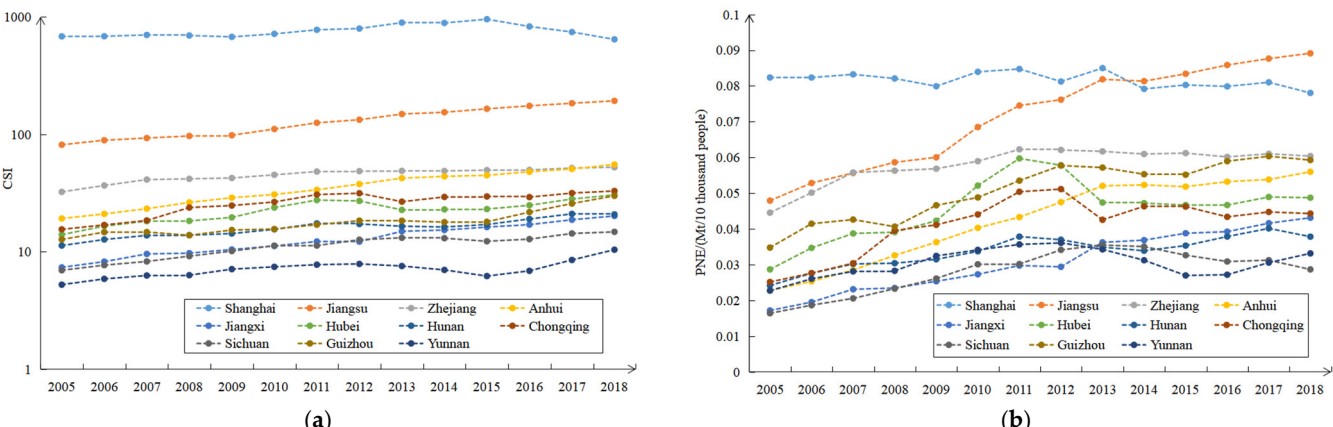

(**a**)                                      (**b**)

**Figure 2.** (**a**) CSI value of 11 provinces and cities in the YREB from 2005 to 2018; (**b**) PNE values of 11 provinces and cities in the YREB from 2005 to 2018 [source: own illustration].

### 3.2. Analysis of the Effect of Urbanization

### 3.2.1. Analysis of the Total Effect

The annual economic urbanization effect, the annual population urbanization effect, and the annual land urbanization effects were extracted by decomposing the *NCE* using the LMDI model. Furthermore, the values of the three effects were added to obtain the annual urbanization effect. The cumulative urbanization impact was determined by aggregating the annual urbanization effects over the study period, using 2005 to 2006 as the base year to demonstrate the pattern of urbanization effect values better. Figure 3 shows the annual impact values and the cumulative effect values. The urbanization shows an inverted U-shape impact on *NCE*, with a shift in the mode of action from facilitation to inhibition. The annual urbanization effect values were positive from 2006 to 2011, except in 2009, when the annual effect value was negative, indicating a significant contribution of urbanization to the growth of *NCE*. During this period, the annual effect of urbanization reached a maximum of 54.62 Mt in 2010. However, urbanization continued to have a negative impact on *NCE* after 2011. The year 2015 witnessed the greatest suppression impact between 2011 and 2018, with a drop in the *NCE* by 100.9 Mt. In terms of cumulative effects, the overall contribution of urbanization to the *NCE* of the YREB throughout the study period was −260.32 Mt. Furthermore, the total contribution value began to be negative in 2014, and the negative values continued to expand until 2018. It is expected that the contribution of the urbanization effect in reducing the *NCE* in the YREB will continue to strengthen in the future.

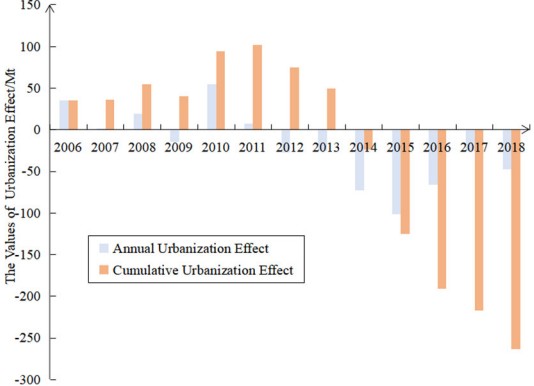

**Figure 3.** Annual and cumulative urbanization effect of the YREB [source: own illustration].

### 3.2.2. Analysis of Each Effect

Figure 4 depicts the annual and cumulative impact values of economic, population, and land urbanization, respectively. The economic urbanization effect is the determining factor of urbanization on the *NCE* of the YREB, which led to a 266.2 Mt reduction in *NCE* from the YREB over the study period. Hence, the total urbanization effect trend is closer to that of the total urbanization effect of economic urbanization. Economic urbanization had an inverted U-shaped impact on the *NCE*, which was boosted initially and later reduced. Economic urbanization promoted a rise in *NCE* from 2005 to 2011, except for 2009, when the annual impact was negative. After 2011, the cumulative impact peaked, and economic urbanization's influence on *NCE* shifted from promotional to suppressive and the intensity of the suppressive effect of economic urbanization on *NCE* was much larger than the promotional effect before 2011. The YREB is one of the most densely populated in China's economic centers and economic hubs of China [50]. Policies and planning at the national level encouraged the transformation and upgrading of the industrial structure of the YREB and bolstered the role of economic urbanization in reducing *NCE*. The transformation and upgrading of China's industrial structure intensified when the 11th Five-Year plan was introduced in 2011. Economic urbanization positively influences *NCE* reduction when the manufacturing industry is upgraded, and the proportion of tertiary industries with low energy consumption and high added value grows [51,52]. Furthermore, from 2013 to 2016, the economic urbanization impact had the strongest driving influence on *NCE* reduction in the YREB. The Chinese government proposed to continue promoting the transformation and upgrading of the industrial structure of the YREB to achieve high-quality economic growth in 2013, and the industrial structure change had a greater inhibitory impact on *NCE* in the YREB after that. The annual effect value of economic urbanization remained negative from 2016 to 2018. The YREB is projected to reduce *NCE* in the future due to industrial structure modification.

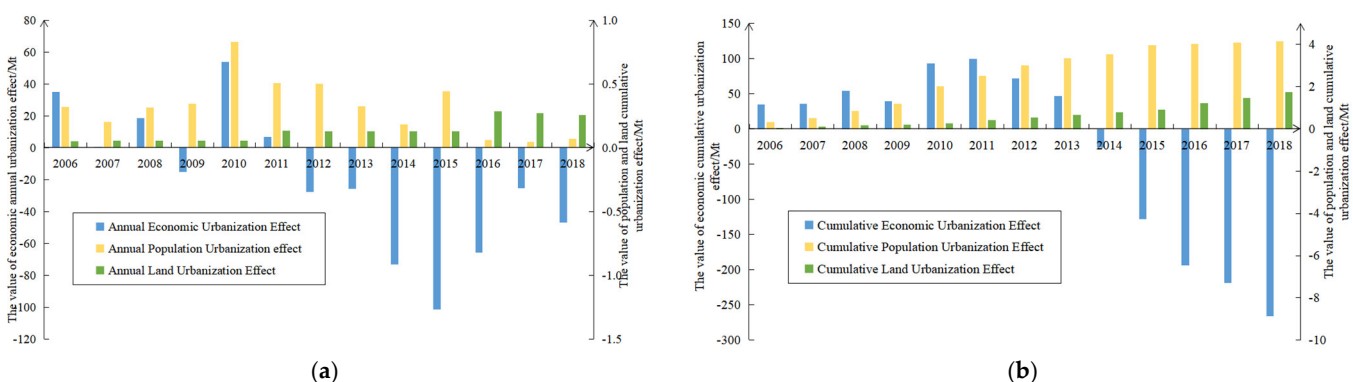

**Figure 4.** (**a**) Annual effect of urbanization in the Yangtze River Economic Belt; (**b**) Cumulative urbanization effect of the Yangtze River Economic Belt [source: own illustration].

The *NCE* of the YREB is promoted by population urbanization. However, since the total energy used by the living sector is significantly lower than that consumed by the production sector, population urbanization has a far lesser impact on regional emissions than economic urbanization. Population urbanization had a cumulative effect of only 4.1469 Mt from 2005 to 2018, roughly 1/60 of the cumulative effect of economic urbanization. From 2007 to 2010, the annual effect value increased steadily, peaked in 2010, and then decreased gradually from 2010 to 2018. Only in 2015 did the impact value grow over the previous year, demonstrating that the positive effect of population urbanization on *NCE* has declined since 2010. The annual effect value of population urbanization to *NCE* was the lowest over the study period from 2016 to 2018, which means that the positive impact of population urbanization on *NCE* in the Yangtze River Economic Belt is expected to deteriorate in the future.

The impact of land urbanization also adds to *NCE*, and this contribution is increasing over time. During the study period, the cumulative land urbanization effect value was 1.7317 Mt. Furthermore, land urbanization in China resulted in a considerable decrease in the area of regional carbon sinks, and an increase in the inhibitory effect on carbon sequestration after the two rounds of five-year plans was implemented in 2011 and 2016. If urban construction continues to deplete carbon sinks in the future, it will have an even greater detrimental effect on regional low-carbon development. However, land restructuring modifications have little influence on regional *NCE*, much less than the impact of economic urbanization.

### 3.2.3. Analysis of Different Regions

The YREB is divided into three regions based on geographical location: the lower reaches (Shanghai, Jiangsu, Zhejiang, and Anhui), the middle reaches (Jiangxi, Hubei, and Hunan), and the upper reaches (Chongqing, Sichuan, Guizhou, and Yunnan) of YREB. The cumulative urbanization effect of all provinces and cities and the lower, middle, and upper ranges of the urban agglomeration were analyzed, and the results are shown in Figure 5. Overall, the cumulative urbanization impact value in the downstream regions was on the decline, while the cumulative effect value in the midstream and upstream regions displayed an inverted U-shaped pattern of increasing and then decreasing. Furthermore, the consequences of urbanization on various regions differed significantly.

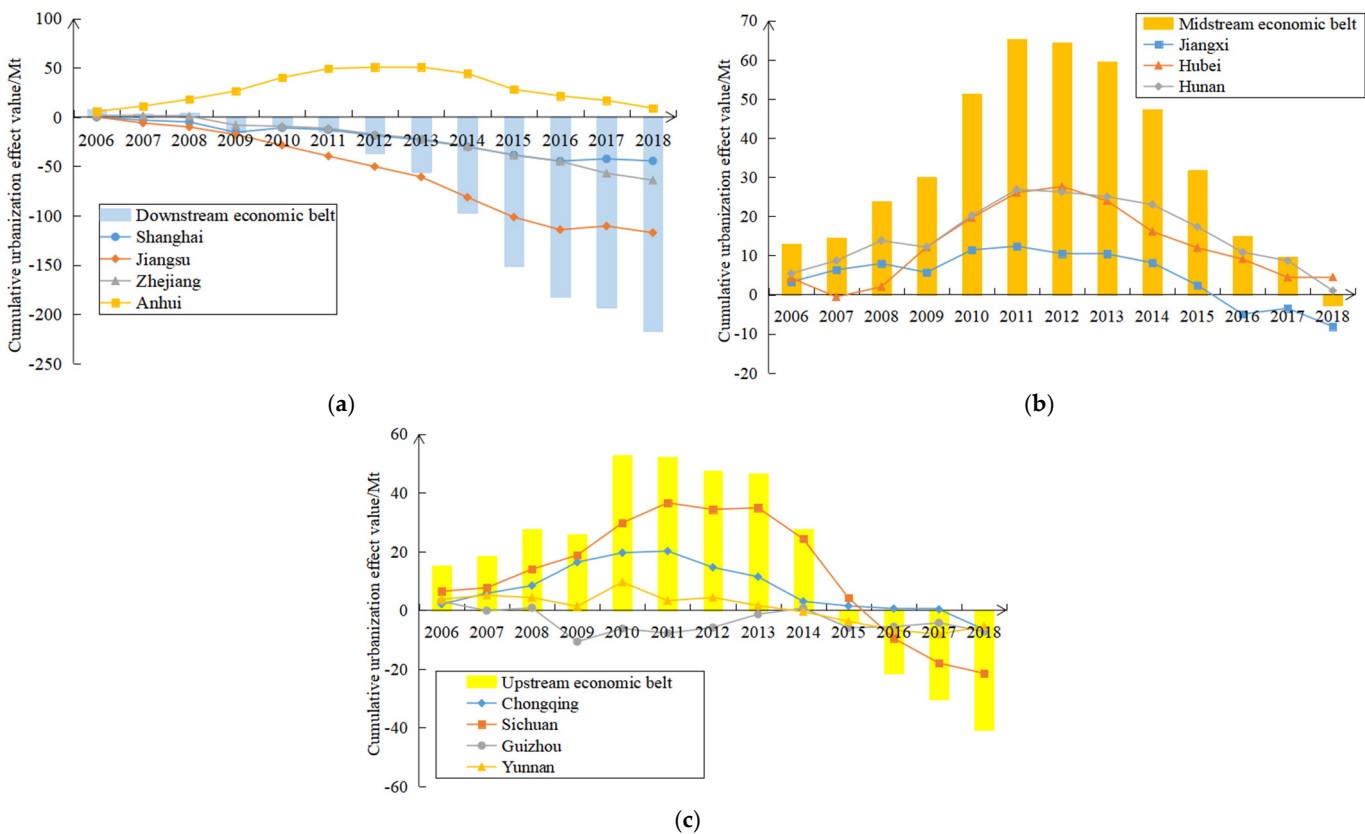

**Figure 5.** (**a**) Cumulative urbanization effect value of provinces and cities in the downstream economic belt; (**b**) Cumulative urbanization effect value of provinces and cities in the middle reaches of the economic belt; (**c**) Cumulative urbanization effect value of provinces and cities in the upstream economic belt [source: own illustration].

From 2006 to 2008, the intensity of urbanization's effect on *NCE* in the lower reaches of YREB was relatively weak. After 2008, the value of the cumulative effect began to decline, indicating that the urbanization effect had reached its peak and was making a sustained

contribution to the low-carbon development downstream [53]. The main cause of this phenomenon was the significant inhibitory influence of economic urbanization downstream from the YREB. Economic urbanization dominated the urbanization effect to the point that it demonstrated a constant inhibitory effect on *NCE* in downstream areas with higher levels of economic development and sophisticated industrial structure. Shanghai, Jiangsu, and Zhejiang generally exhibited comparable changes in urbanization effects, pushing *NCE* reductions between 2005 and 2018. In particular, the cumulative urbanization effect in Jiangsu province reached its maximum decline with the lowest cumulative effect value of −117.33 Mt in 2018. Anhui province was a special case in the provinces and cities in the lower reaches of YREB with an inverted U-shaped relationship between urbanization and *NCE* similar to the midstream and upstream urban clusters. The industrial structure of Anhui Province is relatively backward with relatively weak effects on the decline in *NCE*, as compared to Shanghai, Jiangsu, and Zhejiang. Furthermore, the land urbanization in the Yangtze River Delta urban agglomeration of the lower regions increased *NCE*. Jiangsu province was the province with the largest absolute value of land urbanization effect among 11 provinces and cities, with a cumulative value of 0.28 Mt, with a cumulative value of population urbanization of 2.19 Mt, ranking second only to Sichuan province. Furthermore, in Zhejiang Province, which was the only province in the downstream region where population urbanization had a suppressive effect on *NCE*, the contribution of population urbanization to *NCE* turned from positive to negative.

The urbanization impact exhibited an inverted U-shaped effect on the *NCE* of YREB's middle reaches. The urbanization positively affected the *NCE* of the middle regions from 2005 to 2011, with the cumulative urbanization effect in 2011, after which the cumulative effect value decreased progressively. Additionally, the driving effect of urbanization on reducing the *NCE* in the midstream urban agglomeration was prominent from 2013 to 2016, with a faster decline in the cumulative effect value due to the transformation and upgrading of industrial structures and the improvement of the overall economic level in the midstream region [54]. The variations in the effects in various provinces, including Jiangxi, Hubei, and Hunan, were relatively similar, with an initial rise and subsequent fall indicating the higher level of overall development of the midstream region. Furthermore, the cumulative effect peaks in Jiangxi, Hubei, and Hunan appeared in 2011, 2012, and 2011, respectively, which were very close to each other. However, the absolute value of the cumulative effect in Jiangxi Province was significantly lower as compared to Hubei and Hunan, reflecting relatively less intensity of urbanization in reducing the *NCE* in Jiangxi Province. In terms of different urbanization effects, the population urbanization in Hunan Province had a suppressive effect on *NCE*, while population urbanization in Jiangxi and Hubei had a consistently positive contribution to *NCE* during the study period. Land urbanization also had a promoting impact in Jiangxi, Hubei, and Hunan on *NCE*.

The effect of urbanization on the *NCE* of provinces and cities in the upper reaches of YREB also showed an inverted U-shape during the study period, with insignificant variations in the effects, and the magnitude of the curve was not as large as that in the downstream and midstream areas. The value of the cumulative urbanization effect in the upper reaches of YREB increased from 2005 to 2010, indicating that urbanization was still in its early stages [55]. In the four years after 2010, the cumulative urbanization effect peaked, and the urbanization effect exhibited only a weak suppression of *NCE*. The reduction of the urbanization effect increased dramatically after 2013. There were provincial variations that could be observed in the impact of urbanization on *NCE* in different provinces and cities. The effect curves of Chongqing and Sichuan showed a similar inverted U-shape with the peak year in 2011, with significantly higher effects of the intensity of urbanization effect in Sichuan as compared to Chongqing. In Guizhou and Yunnan, the impacts of urbanization were minimal, with no substantial changes in inverted U-shape, or significant rise and fall. The overall industrial structure of Guizhou and Yunnan was poor, with a slow urbanization process [29] due to their geographical position and economic development level. Therefore, Guizhou and Yunnan should rely on the industrial advantages of the Chengdu-Chongqing

economic circle in the future to promote the transformation and upgrading of industrial structure and economic level while also promoting regional low carbon development. The land urbanization in all provinces and cities in the upstream reaches of YREB increased *NCE*, which is comparable to the downstream and midstream areas. However, population urbanization in Sichuan and Chongqing significantly increased *NCE*, whereas the Yunnan and Guizhou provinces had suppressive impacts. More prominently, the cumulative effect of population urbanization in Sichuan province during the study period was reported to be highest in the YREB, accounting for 2.32 Mt, while the cumulative value in the Guizhou province reached −1.16 Mt. In the future, Sichuan Province should reduce energy consumption in the residential sector, promote a low-carbon lifestyle and encourage a low-carbon development.

## 4. Discussion

Based on the findings of the study, urbanization alters the pattern of carbon sources and sinks through industrial structural changes, urban-rural population migration, and land restructuring, with a multidimensional effect on the regional carbon balance. Economic urbanization dominates the overall urbanization impact on *NCE* among the economic, population, and land effects of urbanization. In terms of specific performance patterns, the impact of economic urbanization on the *NCE* of YREB generally showed an initial promotion, corresponding to the transformation process of the primary industry, which includes agriculture and animal husbandry, to the carbon-intensive secondary industry, including industry and construction, and finally from the secondary industry to the low-carbon tertiary industry, represented by the service industry, in the urbanization process. Although population urbanization contributed to the *NCE* of YREB throughout the research period, its impact was steadily diminishing. In other areas, such as Hunan and Guizhou, the cumulative impact of population urbanization suppressed the *NCE*. The concentration of the urban population makes a great demand on resources in the early stages of urbanization. Low-carbon living is rapidly becoming a trend in urban life as urbanization progresses and inhabitants' living conditions and educational levels rise [27], suggesting an inverted U characteristic similar to that of economic urbanization in the long run due to population urbanization on *NCE*. During the research period, land urbanization limited the expansion of carbon sink areas in all provinces of YREB, resulting in an increase in the scale of net emission. Furthermore, this impact has been rising, implying that the effect of urbanization on $CO_2$ reduction would be overestimated if the variation of regional carbon sink space is ignored.

The impact of urbanization on the regional carbon balance has apparent spatial and temporal heterogeneity due to the combined effect of three major mechanisms. Based on the empirical results of the impact of urbanization on regional *NCE*, the temporal evolution trend exhibits an inverted U pattern, indicating a contributing effect of urbanization on regional *NCE* at the beginning of the study period. However, this effect gradually changed from positive to negative in response to an increase in urbanization level. These findings are consistent with the majority of the available studies [13–15,53]. On the other hand, Liddle and Lung [8] and Wang et al. [56] reported increased emissions in response to urbanization, possibly due to the insignificant effect of urbanization on carbon balance as the study was carried out in the early period of urbanization. Furthermore, the pattern of temporal differences in the impact of urbanization on carbon emissions is mapped onto space due to the different stages of economic and urbanization development in different regions of the YREB. The urbanization effect showed a significant suppression effect on *NCE* in the Yangtze River Delta, which has a high level of urbanization and rapid development, whereas the suppression effect of urbanization on *NCE* was not prominent in the upper region of YREB with relatively low urbanization level. Muhammad et al. [20] and Poumanyvong et el. [57] arrived at similar conclusions by comparing the effect of urbanization on carbon emissions in high-income and low-income countries.

The influence of urbanization on regional *NCE* exhibits spatial and temporal heterogeneity, particularly in the YREB, where the variations in internal development are significant. Being the world's largest river economic belt with the strongest regional influence and synergistic emission reduction in the urbanization process in the YREB has considerable significance for the development of low-carbon urbanization in other large river economic belts.

This can create a low-carbon urbanization plan that takes into account various situations and formulates the low-carbon urbanization strategy based on different regions, classifications, and stages. There are considerable disparities in economic and social development within most large river economic zones. Because urbanization has a nonlinear and inverted U-shaped influence on regional *NCE*, various areas should establish different strategies to reduce *NCE* based on economic, social, and urbanization development. In regions with higher levels of urbanization, efforts should be concentrated on supporting the transition of industrial structure to tertiary industries, increasing the advocacy for low-carbon living, and completely exploiting the effects of urbanization in the reduction in *NCE* [58]. In addition, reasonable land planning policies should be implemented in such areas to optimize the land structure and enhance vegetation and reduce the invasion of land carbon sink space by expanding urban construction land [59]. The government should promote the improvement of production technologies within the secondary industry in regions with low urbanization and poor economic growth. On the one hand, these regions should promote the transformation and upgrading of traditional industries to gradually eliminate enterprises with high energy consumption and high pollution. In addition, the government should accelerate the development of a low-carbon output industrial system and thereby encourage the development of electronic information and other low-carbon industries. This may help to achieve the inflection point of emission reduction more quickly.

Secondly, regional coordinated management are encouraged to be strengthened, and more attention should be paid to the low-carbon urban development system of urban agglomerations and economic belts. The large-river economic zone encourages the establishment of linkage between the upper, middle, and lower reaches of the basin. The backward regions within the economic belt should take advantage of synergistic development between upstream and downstream regions to actively introduce advanced technology and management experience from developed regions and promote the development of local low-carbon technologies. The government should promote city clusters and economic zones to create a city system with large, medium, and small cities having internal cooperation, as well as small and medium cities with relatively close structures and a complete city network with core cities to create a clear division of labor and cooperation and promote energy efficiency.

Thirdly, relying on the advantages of river basin resources, developing clean energy and expanding carbon sink space are feasible approaches to reduce carbon emissions and increase carbon absorption. Large river basins have abundant water resources, with the advantages of hydropower development in the middle and upper reaches, while most of the downstream coastal areas have favorable conditions for developing tidal and wind energy. These advantages might help to reduce fossil energy consumption in urban production and living activities [51]. In addition, large rivers may also offer sufficient water for forest and wetland ecosystems to easily create effective carbon sink areas. The government should actively optimize land use structure, limit land development intensity, regulate the intensity of land development and appropriately expand the proportion of carbon sinks such as forests and grasslands in land planning throughout urbanization to enhance carbon sequestration capacity [60].

## 5. Conclusions

The present study investigated the spatial and temporal patterns of *NCE* based on $CO_2$ emissions from economic and social systems and the $CO_2$ absorption from natural ecosystems in the YREB. The major findings of the study are as follows:

(1) The CSI of the YREB showed an increasing trend during the study period, and the *NCE* increased from 1706.50 Mt to 3106.05 Mt. In this process, the contribution of urbanization to this process followed an inverted U pattern of rise and then decline. Over the research period, urbanization contributed to an overall decline of 260.32 Mt of *NCE* in the YREB.

(2) Economic urbanization dominated the multidimensional effects of urbanization during the research period, displaying an inverted U pattern of development. The population urbanization consistently contributed to the *NCE* of YREB during the study period, with gradual declining trends. In the long term, the impact of population urbanization on *NCE* may resemble economic urbanization by having an inverted U shape. On the other hand, land urbanization consistently promoted the expansion of *NCE* in all provinces of YREB, indicating exaggerated effects with regard to reducing $CO_2$ emissions when the fluctuation in regional carbon is neglected.

(3) The urbanization impact on *NCE* in the YREB has substantial spatial and temporal heterogeneity. From 2005 to 2018, the urbanization impact indicated a considerable reduction of *NCE* in downstream areas with high levels of urbanization and fast growth. However, due to the relatively low level of urbanization in the middle and upper reaches of the YREB, the impact of urbanization on *NCE* initially increased and subsequently declined during the study period, with respective inflection points in 2011 and 2010. In the context of the regional differences in the impact of urbanization on *NCE*, the great river economic belts should develop low-carbon urbanization strategies based on diversities in geography, classifications, and stages. It should also strengthen regional coordination to build a low-carbon urban development system around urban clusters and economic zones.

**Author Contributions:** Conceptualization, X.S. and M.J.; methodology, H.Z.; software, H.Z. and H.X.; validation, X.Y. and G.Z.; data curation, H.Z. and H.X.; writing—original draft preparation, H.Z.; writing—review and editing, X.S., M.J. and X.Y.; visualization, H.Z. and X.Y.; supervision, M.J.; project administration, X.S. and M.J.; funding acquisition, X.S. All authors have read and agreed to the published version of the manuscript.

**Funding:** This article is funded by Key Projects of Jiangsu Planning Office of Philosophy and Social Science (22EYA001).

**Informed Consent Statement:** Not applicable.

**Data Availability Statement:** Not applicable.

**Conflicts of Interest:** The authors declare no conflict of interest.

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
