# Peer review of "Multidimensional Impact of Urbanization Process on Regional Net CO2 Emissions: Taking the Yangtze River Economic Belt as an Example"

_land, doi:10.3390/land11071079_

Round 1
Reviewer 1 Report
Dear Authors,
Congratulations on your valuable research. I would recommend just slight corrections:
Lines 127-129 - 'However, urbanization has resulted in the rapid development of the urban land area, resulting in a large number of woodlands, grasslands, wetlands, and numerous natural water [29-30].' - the citation seems to be wrongly used;
Lines 143 - 145 - 'Scholars have attempted to analyze the causes of these disparities from sev- 143 eral perspectives, including the urban-rural structure of the population and the economic structure. However, only a few studies have included both in the same analytical framework.' - Please, give examples of the scholars, studies (cite them)
Lines 156-157 - 'The Yangtze River Economic Belt (YREB) of China is the world’s largest economy, the most populous town, and the most influential inland river economic belt [36].' - 'The Yangtze River Economic Belt is not a town - try to change the statement (is it properly cited?)
Line 160 - use the word 'centre' instead of 'center'
2.1 Methodology Section - explanations of the various designations in the formula should be described in a legend; the presented descriptions (in the main text) are too long and not clear enough (for example the one in Line 179, Lines 188-193; Line2 207 -221, Lines 227-229)
Line 248 - 'are obtained from the China Statistical Yearbook' - add citation of the yearbook
Lines 252-253 - ' Carbon Emission Accounts & Datasets' - what is it? Could you please add citation or any details of it?
Figures 1, 2, 3, 4, etc. - in brackets add sources of the figures for example [source: own elaboration on the basis of ....]
Figures 2, 3, 4 - enlarge the numbers and the nomenclature shown in the diagrams; the text under and next to the diagrams is not visible
Line 308 - write ' were extracted' instead of 'was extracted'
The literature the Author based on consists of just the chinese literature on the subject. Try to search for some literature from other countries of the world.
Please, take the above mentioned remarks into consideration and include them in the article.
Best wishes with your future projects.
Reviewer
Author Response
Dear reviewer,
Thank you for your valuable comments. We studied them carefully and responded to them one by one (as shown in the attachment). At the same time, we have made corresponding adjustments to the manuscript. Thank you very much again.

Reviewer 2 Report
The rapid development of urbanization is accompanied by a large amount of carbon dioxide emissions. Therefore, the author chooses a good research topic. However, there are still some deficiencies. A few comments for your reference:
(1) The marginal contribution of research needs to be further clarified. The key scientific questions to be addressed and the marginal contribution of the research are unclear.
(2) The mechanism of interaction between core variables needs to be further sorted out. As the author said, the mechanism of urbanization's impact on carbon dioxide emissions is complex, so theoretically, there are several main mechanisms of action that the author needs to make an in-depth review. Otherwise, the whole study feels like a technical paper, not an academic study.
(3) The time period of the study data. Why choose the period from 2005 to 2018, not forward or updated to 2022?
Author Response

(The authors gave the same response as above.)

Round 2
Reviewer 2 Report
I have no other comments.